# Surface- and Tip-Enhanced Raman Scattering in Tribology and Lubricant Detection—A Prospective

**Kun Zhang [1,2], Zongwei Xu [1,\*], Andreas Rosenkranz [3,\*], Ying Song [1], Tao Xue [4] and Fengzhou Fang [1]**

[1] State Key Laboratory of Precision Measuring Technology & Instruments, Centre of Micro/Nano Manufacturing Technology, Tianjin University, Tianjin 300072, China
[2] School of Physics and Electronics, Shandong Normal University, Jinan 250014, China
[3] Department of Chemical Engineering, Biotechnology and Materials, University of Chile, Avenida Beaucheff 851, Santiago 92093, Chile
[4] Analysis and Testing Center, Tianjin University, Tianjin 300072, China
\* Correspondence: zongweixu@tju.edu.cn (Z.X.); arosenkranz@ing.uchile.cl (A.R.);
Tel.: +86-1351-241-2549 (Z.X.)

**Abstract:** Surface-enhanced Raman scattering (SERS) and tip-enhanced Raman scattering (TERS) are fast, convenient, and non-destructive molecular detection techniques, which provide a practical method for studying interfacial reactions with high resolution and accuracy. Both techniques are able to provide quantitative and qualitative information on the chemical properties, conformational changes, order state, and molecular orientation of various surfaces. This paper aims at summarizing the research efforts in the field of SERS and TERS related to tribological systems with a special emphasis on thin film and nanoparticles. This overview starts with a brief introduction for both techniques. Afterwards, it summarizes pros and cons of both techniques related to the advanced characterization of tribologically induced reactions layers. Moreover, the feasibility of both techniques to evaluate the friction and wear performance of new lubricant additives including solid lubricants is discussed. At the end of this review article, the main challenges and future directions in this field are prospected to emphasize the development direction of SERS and TERS in tribology and lubricants.

**Keywords:** surface-enhanced Raman scattering (SERS); tip-enhanced Raman scattering (TERS); tribology; lubricant

## 1. Introduction

Tribology is a multi-disciplinary research discipline, which studies friction, wear, lubrication, and related phenomena of two rubbing surfaces in contact with each other. Therefore, tribology is not only a subject of basic, fundamental research with a rather broad topic range but also an applied problem with strong connections to resource efficiency and energy, thus protecting the ecological environment and improving the quality of life. Nowadays, it has become an important scientific discipline and technical support in many fields of science, technology, and engineering [1–10]. There are certain aspects, which all tribological system/problems have in common. The tribological contact basically consists of two surfaces to which a certain normal load and sliding velocity are applied. The involved friction and wear processes happen in the tribological contact zone. Due to the material pairings involved, this contact zone is normally not accessible for any in-situ characterization technique. In order to explore the underlying friction and wear mechanisms, the worn surfaces need to be characterized after rubbing by different characterization techniques. In this regard, it is essential to investigate the surface roughness, the surface chemistry, and the microstructure of the material [11,12].

For this purpose and depending on the material used, various techniques including white light interferometry and laser scanning microscopy (surface topography), Raman spectroscopy, Sum frequency generation (SFG) spectroscopy, X-ray photoelectron spectroscopy and infrared-spectroscopy (surface chemistry) as well as X-ray diffraction, electron backscatter diffraction and transmission electron microscopy (microstructure) can be utilized. It needs to be emphasized that a multi-method approach is recommendable since only one characterization technique will never be able to reveal the full picture regarding the desired information [12].

Apart from the fact that there is no powerful in-situ characterization technique to assess any of this information during sliding/rubbing, the entire situation is further complicated since friction and wear are influenced by effects on different scales ranging from macro- to nanoscale [13,14]. Regarding the macro-scale, fabrication tolerances and design aspects can be named as potential effects influencing the resulting coefficient of friction and wear. Related to the micron-scale, the surface topography and surface roughness mainly affect the resulting nominal and real contact area [15]. In this regard, it must be emphasized that the real contact happens at the asperity level, which is typically sub-micron and on the nanoscale. The contact at the asperity level typically induces rather high contact pressure. This can lead to pressure-induced reactions thus generating sub-surface reaction layers, which may substantially influence the corresponding tribological properties [16]. These aspects unambiguously underline that the effects occurring at the nanoscale also significantly alter the resulting friction and wear performance. Consequently, the aforementioned surface topography, surface chemistry, and microstructure need to be characterized on different scales [17]. Since SFG uses a polarized light experiment, the orientation of surface molecules can also be deduced by using different polarization combinations of input and output beams [18]. Although SFG is a powerful and highly versatile spectral tool for identifying surface molecular species and providing information about surface structure, the characterization of this method on the nanoscale can be considered as time-consuming, costly, and rather sophisticated [19–21]. In this sense, it would be highly appreciated to provide fast and easily applicable techniques, allowing for a reproducible characterization of the surface chemistry with high resolution and accuracy.

In this regard, far-field Raman spectroscopy (FFRS) seems to be perfectly suited to investigate the surface chemistry of worn surfaces, thus gaining a deeper understanding about the involved friction and wear mechanisms. The Raman scattering effect is caused by the coupling of the dipole moment induced by an electromagnetic wave and the vibration quantum of the target analyte. FFRS offers a wide range of advantages since it provides fast, simple, repeatable, and more importantly, non-destructive qualitative and quantitative analysis, which can be used to characterize the molecular structure and conformation information [22]. This shows that Raman scattering can be utilized as an in-situ detection technique for tribo-chemical reactions under wear conditions as already shown for tribo-chemical reactions in case of stearic acid crystals [23–25]. FFRS has a significant drawback since the probed information is on the order of 1 micron. Therefore, FFRS is rather insensitive for surface species and thin, nanometric films. In order to overcome this shortcoming, near field techniques such as surface-enhanced Raman scattering (SERS) or tip-enhanced Raman scattering (TERS) can be utilized. Both techniques help to overcome the shortcoming of a low superficial sensitivity of FFRS and to obtain structural information, which cannot be easily obtained by FFRS.

## 2. Fundamental Mechanisms

### 2.1. Fundamental Principle of SERS

SERS is widely used in surface and interface adsorption, orientation and configuration of biomolecules, conformational studies, and structure analysis among others. It can effectively analyze the adsorption orientation, adsorption state, and interface information of compounds on the interface, etc. At present, there are two main theories describing the underlying SERS mechanism, which are based upon chemical and physical enhancement. The physical enhancement mechanism includes the surface electromagnetic field model, the antenna resonance sub-model, and the surface mirror

image field model. Gold, silver, and other noble metals showed superior SERS ability, and even single molecule detection was realized through local surface plasma resonance (LSPR, the so-called "hot spot") on the surface of metal nanostructures [26,27]. LSPR, which is the fundamental mechanism of SERS, occurs when the collective oscillation of valence electrons in metal nanoparticles resonates with the frequency of the incident light [28–30]. A schematic illustration of LSPR is shown in Figure 1. The plasma resonance frequency can be modified by controlling the geometry of individual gold and silver nanoparticles during preparation and synthesis. Currently, photolithography, focused ion beam microscopy, atomic layer deposition, and Langmuir–Blodgett thin film transfer can effectively control these nanostructures [31,32]. Furthermore, SERS can be considered as a powerful analytical tool, which is useful for sensitive and selective detection of molecules adsorbed on the surface of nanostructured (i.e., rough) metal.

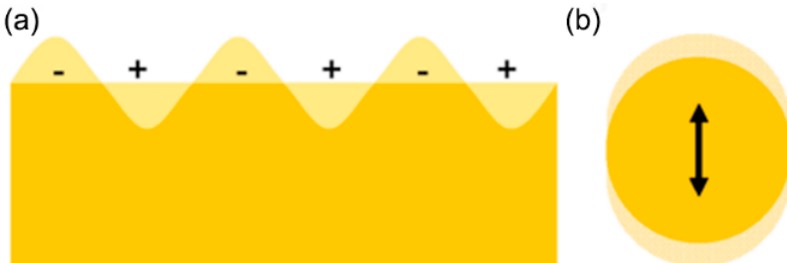

**Figure 1.** Schematic illustration of (**a**) surface plasmons and (**b**) a localized surface plasmon. (Adapted with permission from ref. [33]).

*2.2. Fundamental Principle of TERS*

In TERS, a Raman spectrometer is combined with a scanning device thus making use of localized surface plasmon phenomena to induce near-field effects and to probe small surface areas [29,34–38]. With regard to the scanning device, this can either be an atomic force microscope (AFM), scanning near field optical microscope (SNOM), a shear force microscope (SFM), or a scanning tunneling microscope (STM) [37,38]. Figure 2 shows a schematic diagram of different TERS setups using a STM, AFM, SFM, and SNOM. The laser light of the Raman spectrometer is focused on the metalized tip of any of those scanning devices. If the tip is in close proximity of the surface of interest, the metalized tip acts as an antenna for light, thus inducing surface plasmons and, consequently, enhancing the incident and emitted light fields [35,39,40]. As can be seen in Figure 2f, different arrangements (bottom, side, and top) have to be distinguished related to way how to excite the sample and to collect the resulting signal.

In order to explain the underlying principle of TERS in a better way as well as to unravel the importance of the tip distance in TERS experiments, Figure 3 plots the recorded Raman spectra measured with a top-illuminating TERS system for three distances. The probed material was a film of carbon onions (thickness of about 1 μm) deposited by electrophoretic deposition on a Silicon substrate.

As a first step, the tip was completely retraced until reaching a distance of about 50 micrometers between the AFM tip and the sample to be measured. Afterwards, a Raman spectrum was recorded, which shows the typical D- and G-peak for carbon nanoparticles. This spectrum (blue color) can be assigned to the far-field Raman spectrum and reflects a probed area on the order to μm$^2$. After having recorded the initial far-field spectrum, the distance between tip and sample was continuously reduced until recording the black curve, which reflects a distance of about 1–2 nm and is therefore named "in contact". The black spectrum reflects the tip-enhanced signal. Although significantly reducing the probed area, a pronounced enhancement in the D- and G-peak can be seen. Taking the highly reduced probed area into consideration, enhancement factors on the order to $10^5$–$10^6$ can be estimated, which shows the tremendous potential to probe thin carbon films/nanoparticles.

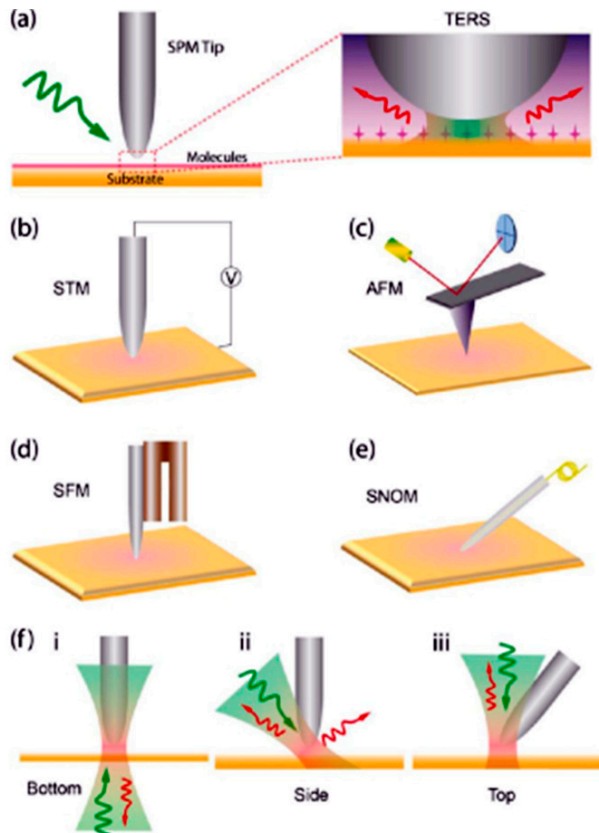

**Figure 2.** (**a**) Schematic illustration of different TERS setups using a STM (**b**), AFM (**c**), SFM (**d**), and SNOM (**e**). (**f**) Three different modes of excitation and collection need to be distinguished: (i) bottom excitation and collection, (ii) side excitation and collection, and (iii) top excitation and collection. (Adapted with permission from ref. [38]).

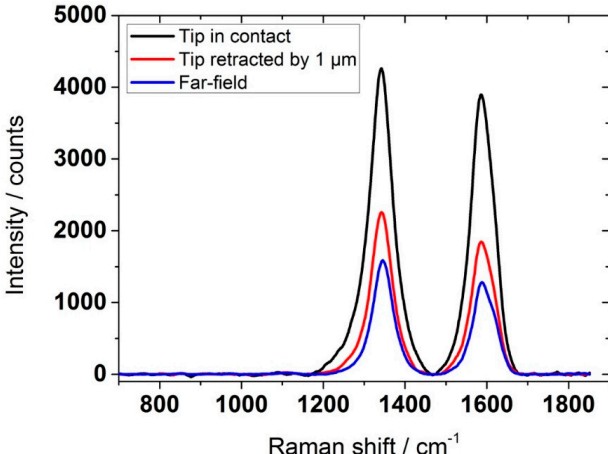

**Figure 3.** Comparison of three different tip-enhanced Raman spectra, all recorded with the same top-illuminating TERS setup but with a different tip distance. The probed material was a film consisting of carbon onions. All recorded spectra were baseline corrected. (Unpublished data of A. Rosenkranz).

Currently, SERS is used in different fundamental research areas including physical chemistry, materials science, surface science, nanoscience, biomedicine, environmental monitoring, food safety and archaeological research [41–50]. In recent years, SERS has been used to detect single molecular chains and single nanoparticles, which underlines its resolution and potential [51,52]. Additionally, SERS has also been widely utilized in catalysis, lubrication, adhesion, corrosion inhibition, and other

fields [53–56]. SERS provides a new approach to investigate the underlying tribo-chemistry in friction and wear processes as well as to design/develop advanced lubricants, which has attracted great attention in the scientific community. The study of friction and lubrication, as well as their relationship, is of great significance to the failure analysis of friction pairs, the evaluation of lubrication efficiency, the extension of the service life of equipment, and the improvement of the reliability of machine components. This paper aims at summarizing the usage of SERS and TERS related to tribological problems and questions thus discussing potential advantages and shortcomings of both techniques related to the advanced characterization of worn surfaces.

## 3. Application of SERS and TERS in Tribology and Lubricant Detection

### 3.1. Application of SERS in Interface Tribology Reaction Detection

Tribology aims at understanding the microscopic dynamic behavior and changes occurring at the interface between two rubbing surfaces under a certain load and velocity. As outlined in the introduction, tribological properties are affected by contributions coming from different scales including the atomistic, nano-, micro- and macroscale. It is well known that exactly these interfacial processes are the key aspects to reduce friction and wear. It has been recognized that research in the field of nano- or submicron-sized materials/particles used as lubricant additives or solid lubricants is promising to achieve low friction and wear. Raman scattering spectroscopy is a good tool to study the chemical structure of materials. However, due to its low sensitivity and micron depth resolution, it is difficult to analyze ultra-thin films, surfaces, or buried interfaces. SERS can make up this shortcoming of low scattering cross section of far-field Raman spectroscopy in the detection process of thin lubricating films [29].

Honda et al. studied the tribological properties of ultra-thin carbon, Ag, and H films (monolayers) on Si (111) substrates, thus addressing the tribological effects of surface atoms [57]. It has been verified that the friction force in these sliding systems was extremely small under ultra-high vacuum conditions. They have also demonstrated that the chemical surface properties have a great impact on friction. Particularly, Ag monolayers showed a highly reduced COF and can be also used for SERS due to their thickness, thus bearing the potential to further explain the excellent frictional properties observed. Hashigushi et al. have utilized SERS to estimate the stress in strained, thin Si films applied on relaxed SiGe located on oxidized Si substrates [58]. For unmodified Si substrates, no Raman peak shift has been observed. In contrast, for the strained, thin films, a significant peak shift towards higher wavenumbers was observed. Comparing both results, a detailed stress analysis became possible.

The double bond of unsaturated vegetable oils is widely believed to be responsible for the oxidation sensitivity of these oils. It has been initially expected that the film growth may be related to the degree of saturation, since highly unsaturated oils are theoretically more thermally unstable, thus having a greater potential for film growth. Chua et al. used SERS to study tribo-chemically induced boundary films formed by refined and unrefined rapeseed oil (Figure 4) [59]. To the best of our knowledge, this was the first time that SERS has been used to study surface films on metal surface lubricated by vegetable oil. In the SERS spectrum of refined rapeseed oil, the strongest peaks appeared near 235, 940, 1377, and 2930 cm$^{-1}$, while the peak at 940 cm$^{-1}$ is a double peak of 930 and 955 cm$^{-1}$. The SERS spectra of unrefined rapeseed oil showed strong peaks at 215, 930, and 1395 cm$^{-1}$, which correspond to $\nu$(Ag–O), $\nu$(C–COO–), and $\nu$(COO–), respectively. The assignment of the peaks identified from the spectra of both canola oils is summarized in Table 1. These results provide strong evidence for changes in fatty acids as they slip. In terms of the chemical properties and surface adsorption characteristics of the boundary layer formed by tribo-chemistry, SERS can provide important information.

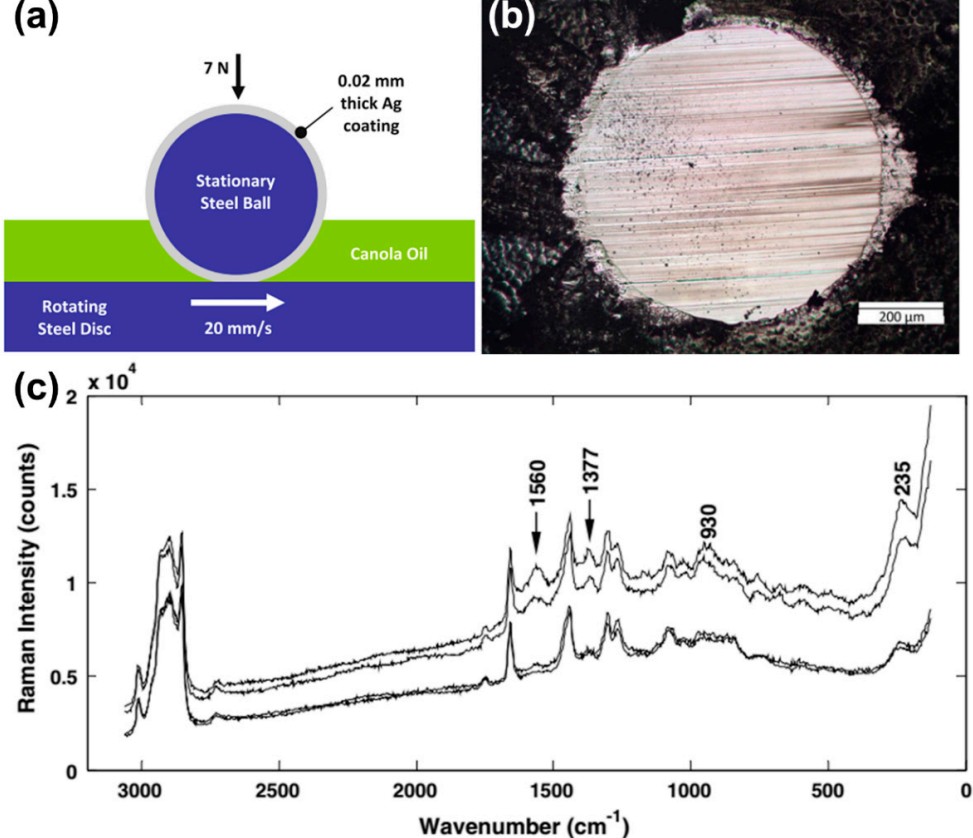

**Figure 4.** (**a**) Illustration of the ball-on-disc contact under fully flooded lubricated conditions. (**b**) Optical microscope image of the unwashed wear scar of the silver-coated ball for unrefined canola oil. (**c**) SERS spectra of the unwashed ball wear scar of refined canola oil with enhanced vibrational bands (labeled). (Adapted with permission from ref. [59]).

**Table 1.** Assignment of the SERS peaks for refined and unrefined canola oil. (Adapted with permission from ref. [59]).

| Vibrational Band Assignment | Peak Position (cm$^{-1}$) | |
|:---:|:---:|:---:|
| | Refined Canola Oil | Unrefined Canola Oil |
| $\nu$(Ag–O) | 235 | 215 |
| CH3 rock, $\tau$(C–H) | 760 | 760 |
| $\nu$(C–C) | | 892 |
| $\nu$(C–COO-) | 930 / 955 | 930 |
| $\delta$(C–H), $\nu$(C–O) | 1030 | |
| $\nu$(C–C)G, $\nu$(PO$^{3-}_4$) | | 1090 |
| $\delta$(C–H), $\nu$(C–C) | 1167 | |
| $\tau$(CH2) | 1300 | 1295 |
| $\nu$(COO-) | 1377 | 1395 |
| $\delta$(CH2) | | 1439 |
| $\nu$(C=C) aromatic ring | 1560 | |
| $\nu$(C=C) olefinic chain | 1630 | 1630 |
| $\nu$(C=O) | 1700 | |
| $\nu$(CH2)S | | 2855 |
| $\nu$(CH2)A | | 2875 |
| $\nu$(CH3) | 2930 | 2930 |

$\nu$ stretching, $\tau$ twisting/wagging, $\delta$ bending/scissoring, $_S$ symmetric, $_A$ antisymmetric, $_G$ gauche, $_T$ trans.

Surface films (single- and multilayers) formed by the adsorption of polar additives play an important role in reducing friction and wear. Raman scattering can provide an in-situ detection technique for tribo-chemical reactions under wear conditions, such as of stearic acids on copper surfaces [23]. The Raman scattering on the disk surface shows no chemisorbed hexadecane since there is no frequency shift in the hexadecane fingerprint (Figure 5). The strong band at 629 cm$^{-1}$ indicates a chemical interaction between the stearic acid and the copper surface. After a sliding distance of 100 m, the stearic acid and the copper surface form a chemical complex, which plays an essential role in protecting the surface from severe wear. The characteristic spectral lines at 491 cm$^{-1}$ of copper oxide are unique products of tribo-chemical reactions (Figure 5d). No cupric oxide will be formed on the copper surface in thermal oxidation reactions below 200 °C in air [60]. Thus, intermolecular reactions of the compound form copper stearate. After a sliding distance of 400 m, the Raman spectra of copper surface show peaks, which can be associated to carboxyl group at 1540 cm$^{-1}$ and Cu-O band at 243, 288, and 623 cm$^{-1}$. Using SERS, the main differences between copper stearate and stearic acid were the displacement of the Cu-O band and the C=O band.

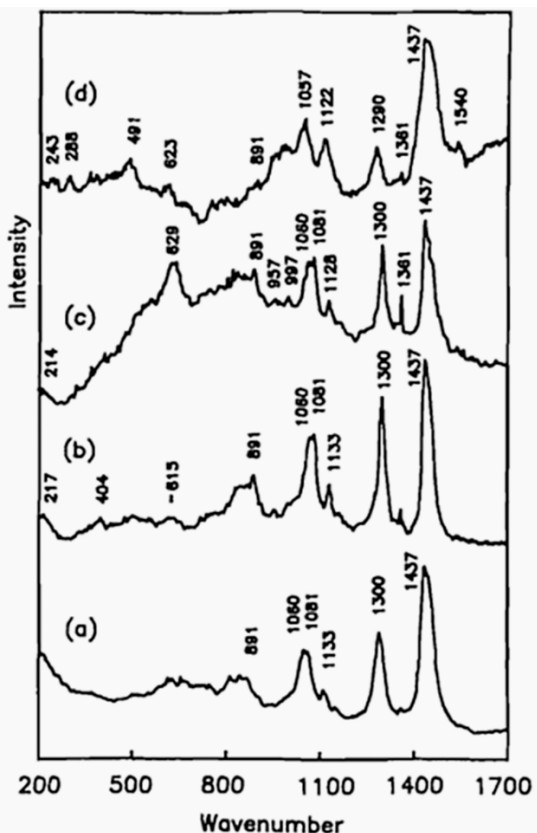

**Figure 5.** Raman spectra of (**a**) hexadecane on copper surface prior to rubbing, (**b**) after rubbing (100 m), (**c**) lubricated with hexadecane containing 0.4% stearic acid after rubbing for 100 m sliding distance, and (**d**) after 400 m sliding distance. (Adapted with permission from ref. [23]).

As a common additive in processing, stearic acid can not only form an electrical insulation coating on the surface of silver but also prevent flake agglomeration. Furthermore, the interaction between silver flakes and the lubricating oil may affect the interaction between the flakes and the base metal to some extent. Vibration analysis of stearic acid adsorbed layer on silver substrates was reported by Miragliotta et al. [61]. By performing SERS as a function of the sample´s temperature, the chemical properties of the interface system at different temperatures were studied. The effect of temperature on carboxylate and silver SERS response in the C-H stretch zone was investigated. Since the carboxyl group coupled through an oxygen atom with the Ag substrate, the SERS spectrum at room temperature

showed the peak of the carboxylate stretching vibration. Analyzing the vibration amplitude and position of the peaks observed in the SERS spectrum, it can be concluded that the SERS spectrums of stearate layer have almost no changes in a temperature range from room temperature up to 140 °C. When the temperature exceeds a temperature of 150 °C, the carboxylate layer is partially decomposed, and an amorphous carbon layer starts to form on the silver surface. This leads to the formation of three new peaks located at 1405, 1597, and 2950 cm$^{-1}$, which goes hand in hand with a continuous decrease of the stearate peaks in the spectrum. The new bands were identified as basic (1405 and 1597 cm$^{-1}$) and combinatorial peaks (2950 cm$^{-1}$) of amorphous carbon. It is worth to mention that the SERS results showed that C-H tensile vibration still existed on the Ag surface at 175 °C. This indicates that the surface layer of stearate has not been completely decomposed.

The generation of transfer films is a common phenomenon in tribology when using solid lubricants. Krick et al. used SERS to characterize the material transfer in-situ when rubbing a gold coated prism against different solid lubricants including polytetrafluoroethylene (PTFE), ultra-high-molecular-weight polyethylene (UHMWPE) and graphite. In this regard, SERS unambiguously helped to identify the chemical species present in the transferred films [62]. The observed spectra of PTFE and heated PTFE transfer films are basically consistent with those of the bulk PTFE (Figure 6). This can be compared to the far-field spectrum of the PTFE transfer film generated by sliding PTFE on quartz (without gold layer). The surface-enhanced Raman spectrum similar to the PTFE reference spectrum was obtained by detecting the PTFE transfer film formed on the gold-plated quartz. However, the detection of dense PTFE on quartz has no significant Raman enhancement. They also found that when UHMWPE slides over the gold sensitive layer, the surface reflectance of the SPR signal changes little. In contrast, when the graphite sample slides over the gold sensitive layer, the SPR signal monotonically increases.

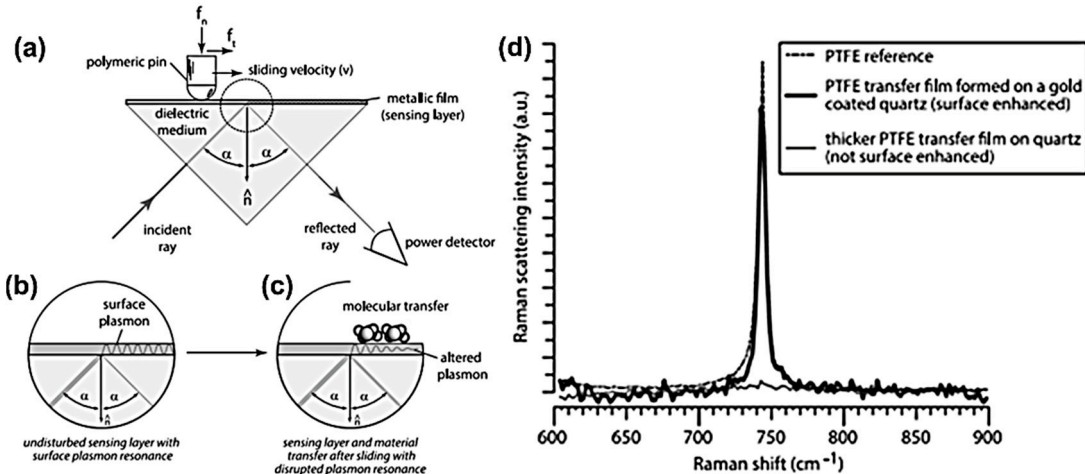

**Figure 6.** In-situ surface plasmon resonance tribometry. (**a**) Illustration of the in-situ surface plasmon resonance tribometer that slides a sample across a metallic sensing layer on a quartz prism utilizing the Kretschmann configuration. (**b**) Sketch of the Kretschmann configuration for the generation of surface plasmons. (**c**) Schematic of the modification of the surface plasmon due to molecular adsorbates or transfer on the metallic sensing layer. (**d**) SERS spectra for PTFE transfer film formed during in-situ experiments. A PTFE reference spectrum and a spectrum for PTFE transferred to glass by sliding are shown for comparison. (Adapted with permission from ref. [62]).

In this section, SERS has been reviewed regarding its ability to detect ultra-thin transfer films on different surfaces induced by tribo-chemical reactions. This brief summary shows that SERS is a powerful tool to characterize these thin layers thus providing more fundamental knowledge about the involved friction and wear processes. Especially the study presented by Krick et al. underlines the interesting possibilities of in-situ SERS setups to further characterize the reaction layers induced by thin solid lubricants.

### 3.2. Application of SERS in Lubricant Detection

The properties and applications of metal or metal oxide colloidal particles depend on their size, stability, and concentration in colloidal solutions. Low particle content and poor stability of concentrated colloids are the main limiting factors in various application fields such as photocatalysis, heat transfer, and lubrication.

Silver nanoparticles are a frequently used nanomaterial. In terms of wear reduction, silver nanoparticles become a lubricant additive with increasing research interest. Ghaednia et al. investigated the tribological properties of polyethylene glycol lubricants with nano-silver [63]. Tu et al. introduced a new strategy of using (3-aminopropyl) trimethoxysilane (APS) a surface modifier and reducing agent to prepare high-sensitivity silver nanoparticles for SERS substrates in situ on silicon wafers. This work makes a great contribution to the generation and distribution of silver nanoparticles and provides a new tool to clarify the mechanism of liquid super lubricating systems through real-time interface state tracking [64]. Their SERS substrates combine high sensitivity with excellent uniformity. Glycerol plays an important role in liquid super lubrication systems [65–67]. SERS substrates with higher sensitivity and good uniformity exhibit excellent performance in detecting glycerol. The SERS substrate with high sensitivity was obtained by using APS as a surface modifier and reducing agent in situ to prepare silver nanoparticles on silicon wafers, showing excellent performance in detecting glycerol, which opens a new window for real-time tracking of friction interface state.

2,5-dimerhydryl-1,3,4-thiadiazole (DMTD) can be used as a lubricant additive aiming at preventing corrosion and reducing friction [68]. Huang et al. studied the mechanism of DMTD on the copper surface using SERS [69]. They found that DMTD and Cu ions have the same coordination mode in two compounds formed by external cyclic sulfhydryl group, and the difference in band position may be caused by the different oxidation states of copper ions.

Raman spectroscopy is well suited to study carbon overcoats (COC) in magnetic disks. However, because of its limited sensitivity and micrometer depth resolution, it is difficult to analyze these ultra-thin films with conventional far-field Raman spectroscopy. Yanagisawa et al. observed the layered structure of highly oriented pyrolysis graphite (HOPG) by analyzing the chemical structure from the surface to the buried interface of ultra-thin films with a depth resolution of 0.1 nm [70]. They also studied the properties of layered thin films consisting of diamond-like carbon (DLC), perfluorinated polyether (PFPE) and phosphoronitrile derivative (A2OH). The results show that the displacement of phenyl wavenumber is lower when functional groups are adsorbed on DLC surface. Compared with the deeper position, a larger intensity ratio ($I_D/I_G$) between the D peak and G peak was observed around the thin film surface, indicating that the surface of DLC film was defective. The results also demonstrated that the DLC films prepared by chemical vapor deposition contained many organic constituents. Additionally, Yanagisawa et al. developed a plasma SERS sensor, which can acquire Raman spectra with an improved S/N ratio [71]. Ultra-thin DLC, nitride DLC and lubricated DLC films on hard disk drives were analyzed using the organic molecular sensor with plasma antenna and cross-correlated with DFT calculations [72]. Due to the improved SNR of this approach, the Raman spectrum of a DLC film with a thickness of 0.4 nm and a lubricated DLC film were obtained. Moreover, it was shown that chemical interactions between phenyl groups and the DLC film are inversely proportional to the nitrogen content. Additionally, chemical interactions between hydroxyl groups and nitrogen-containing DLC films can be enhanced as well.

Carbon overcoats and lubricating oil films can suffer from thermal damage and mechanical degradation problems at high temperatures and mechanical loads [73,74]. Yanagisawa et al. used a plasma SERS sensor to observe the heating behavior of lubricating oil films on carbon overcoats used in hard disk drives [75]. The spectral changes of lubricating oil film during laser heating were measured, and the spectral changes of lubricating oil film were calculated using the Stokes/anti-Stokes intensity ratio. They found that the lubricating oil film composed of PFPE evaporates at more than 290 °C (Figure 7a), which is in good agreement with the results of thermogravimetric analysis. The distance between the lubricating oil film and SERS sensor varied between 0 (contact) and 50 nm (Figure 7b).

The study also found that when the laser power reached 10 mW or more, the lubricating oil film was evaporated.

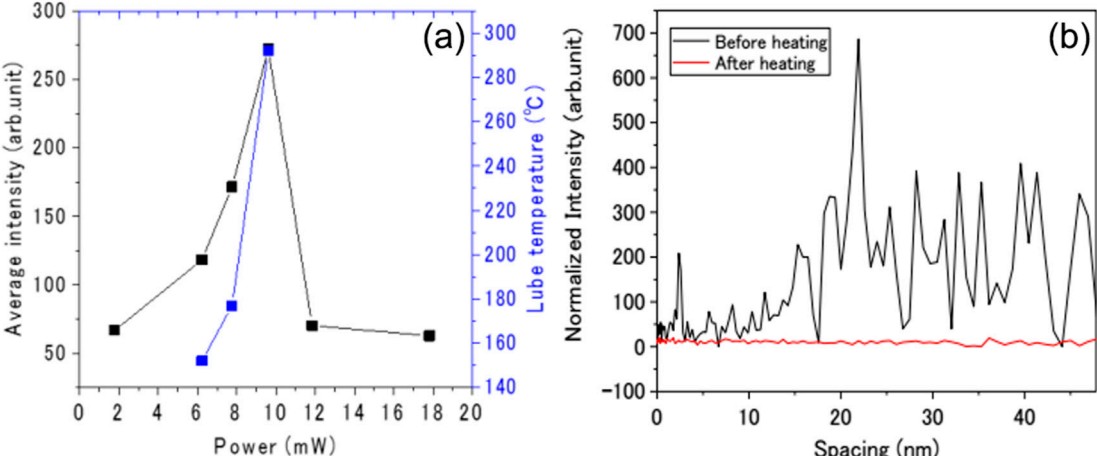

**Figure 7.** (**a**) Intensity change of the ether group and the temperature for lubricant film as a function of the laser power. (**b**) Intensity change of the ether group as a function of the spacing prior to and after laser heating. (Adapted with permission from ref. [75]).

Under sliding friction, it can be assumed that the arrangement of surface molecules may be quite different from that in the body. Ge et al. studied the molecular orientation and conformation changes of the surface of polyimide colligation layer by using surface layer decoration (SLD), SERS, and optical second harmonic generation (SHG) experiments [76]. In this paper, silver colloidal aggregates were deposited directly on the surface of 6FDA-6CBBP films without friction, and strong enhanced Raman signals of 6FDA-6CBBP films prior to and after friction were detected (Figure 8). It can be deduced that the cyan biphenyl on the side chain has an inclined conformation relative to the surface substrate after friction, which is significantly different from the near-plane conformation prior to friction. The SERS results showed that the overhanging cyanobiphenyls in the side chain and skeleton had chain conformation on the polyimide surface similar to the plane prior to rubbing. Friction not only causes the main chain to tilt, such as the imine-benzene structure but also causes the vertical side chain to reposition significantly on the surface. SERS can recognize the molecular conformation near the plane of the skeleton on the surface of the original polyimide and in the vertical orientation.

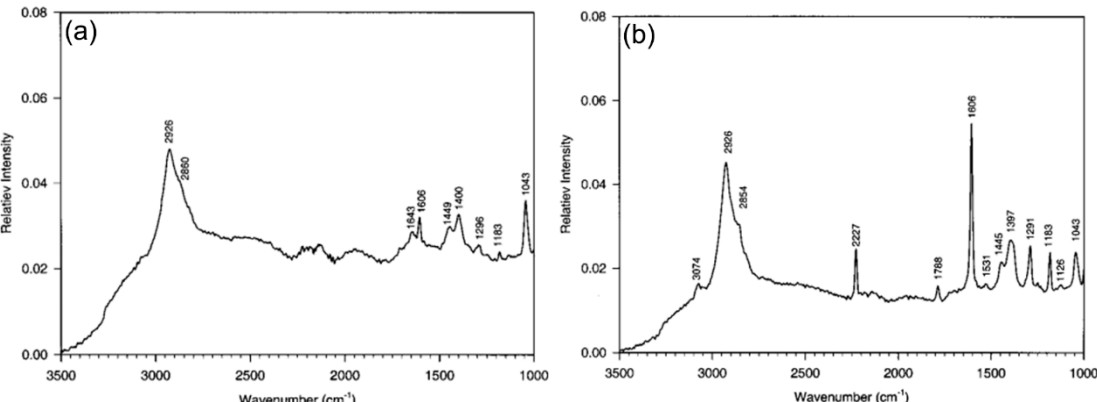

**Figure 8.** SERS spectrum of 6FDA-6CBBP from an unworn thin film contacted with a metallic layer. SERS spectrum of 6FDA-6CBBP from a worn thin film contacted with a metallic layer. (Adapted with permission from ref. [76]).

In conclusion, this section mainly introduced different studies on lubricants and related aspects based upon SERS detection and proved the potential of SERS related to the detection of lubricant

additives and their chemical, structural, and conformal changes. Especially the research conducted on carbon overcoats shows the tremendous potential of SERS to explore the underlying friction and wear mechanisms in carbon systems. Having in mind that carbon nanoparticles such as graphene, carbon nanotubes, carbon onions among others are intensively applied as solid lubricants, the usage of SERS in these systems can pave the way for more efficient tribological systems.

### 3.3. TERS as a Technique with High Chemical Resolution

TERS has been utilized for sensing and chemical imaging of the nanoscale using carbon nanotubes (CNTs) as probed material. Due to the huge field enhancement, also materials that are normally known as a weak Raman scattering can be probed without any problem. Hayazawa et al. investigated single-wall CNTs by TERS and compared the respective far-field spectrum with the enhanced signal. For single-wall CNTs, they demonstrated a selective enhancement of the G-peak. In addition, they also studied amorphous carbon by TERS. Amorphous carbon revealed a more pronounced D-peak though. For the radial breathing mode, which is located at around 160 cm$^{-1}$, a pronounced band splitting was observed in TERS [77]. Chen et al. verified a maximum resolution of 1.7 nm using TERS based upon a STM and individual CNTs as material of interest. Local defects, bundling effects and different types of CNTs were imaged with high resolution using this setup. Those authors also presented a stronger enhancement of the G-peak compared to the D-peak [78]. Chaunchaiyakul et al. used a scanning electron microscope combined with a Raman spectrometer to study multi-wall CNTs. Regarding the D- and G-peak, they reported about a strong and selective enhancement of the G-peak compared to the D-peak. The intensity ratio ($I_D/I_G$) and the area integrated ratio ($A_D/A_G$) showed a step-like shape, which was correlated to the number of walls present in the multi-wall CNTs [79]. Using a TERS based upon a STM, Liao et al. verified a spatial resolution of 0.7 nm for individual CNTs. The high resolution makes the tracking of strain-induced structural changed possible TERS. Bending experiments on individual CNTs and in-situ TERS measurements revealed a split-up of the G-peak. This band splitting strongly depended on the degree of deformation. With this set-up, even structural changes on the inner and outer diameter of CNTs were identified [80]. Saito et al. studied the influence of the polarization of the exciting laser light on the resulting spectra of single-wall CNTs by TERS. The polarization was varied between s-polarization (parallel to the sample plane) and p-polarization (perpendicular to the sample plane). They observed a selective enhancement of the G-peak using s-polarized laser light, while p-polarized light enhances more the radial breathing mode. This selective enhancement of vibrational modes can be explained by the interrelation of the laser polarization and the orientation of vibrational moments on the molecular level. Comparing far- and near-field measurements, a maximum enhancement factor of 2.5 could be observed for the G-peak [81]. Moreover, Rosenkranz et al. have used TERS to study the structural transformation of nanodiamonds to carbon onions for different annealing temperatures. In this context, far-field Raman spectroscopy does not provide any Raman signal for nanodiamonds due to the sp$^3$-hybridization of these nanoparticles. Due to the significant field enhancement in TERS, even nanodiamonds can be measured and analyzed. This tremendously underlines the potential of TERS to probe materials with a low scattering cross-section, which are difficult to measure under far-field conditions. Regarding the structural changes in the transition from nanodiamonds to carbon onions, FFRS, and TERS shows similar trends, thus confirming the ongoing structural changes at the nanoscale [82]. Moreover, Rosenkranz et al. used the same TERS setup to probe ultra-thin DLC coatings commercially used on hard disk drives. These coatings have an estimated thickness of 2–3 nanometers. Due to the limited thickness, these coatings are almost not accessible via FFRS. Using TERS, Rosenkranz et al. reproducibly measured the Raman spectrum of these ultra-thin carbon coatings with clear D- and G-peaks [83]. This underlines that TERS makes the analysis of ultra-thin coatings accessible, which opens new doors in the characterization and understanding of these systems.

*3.4. Challenges and Opportunities*

Nowadays, there are different challenges that need to be overcome to make SERS and TERS suitable and applicable to in-situ tribology, thus probing chemical reactions at the frictional surface/interface and studying chemical and lubricating properties of formed tribo-films/layers by tribo-chemical reactions. (1) From the perspective of tribology, a large amount of energy is converted into heat during sliding, thus increasing the interfacial temperature. To assess these interfacial temperatures can be considered as a great challenge, which may be solved with the analysis of Stokes and Anti-Stokes contributions. (2) Different chemical reactions, including such as polymerization, crystallization, or irreversible degradation may happen in the interface during sliding thus leading to material degradation or the formation of wear debris. These processes are highly dynamic in nature and therefore ask for real-time monitoring. In order to realize that, improved setups combining basic tribometers with SERS and TERS need to be developed to allow for an in-situ real-time monitoring. The physical and chemical changes between sliding surfaces or between sliding surfaces and the environment can be better understood by real-time detection of chemical changes during friction. (3) The applicability and reliability of SERS and TERS need to be improved in different aspects. Related to SERS, reproducible SERS-active films need to be easily fabricated and applied to the tribological interface. Regarding TERS, the fabrication of TERS-active tips needs to be improved significantly. Commercially available tips may not work with every substrate or are optimized for certain substances. Therefore, reproducible methods for a reliable tip fabrication with high sensitivity are urgently needed to assess the full potential of this technique. In this context, the shape of the nanoparticles at the tip may be optimized in order to even achieve higher field enhancements. (4) SERS and TERS are surface-sensitive methods. Especially TERS shows a rapidly decreasing sensitivity in the first few nanometers. Therefore, the position of the respective material to be probed in the tribofilm is highly important. If the tribofilm is too thick, you may not capture all the respective processes going on, which can lead to wrong interpretations of the respective signals.

## 4. Summary and Perspective

Tribology is a highly complex, multi-disciplinary research discipline related to surface science and surface engineering. A key aspect in this research field is the detailed knowledge about the underlying friction and wear mechanism, which underlines the need for more real-time and in-situ characterization techniques capable to detect structural, chemical, and microstructural changes on small scales (submicron or even nano). In this review, the research progress of SERS and TERS in surface engineering, biological tribology, magnetic recording, and tribology has been summarized. Enhanced Raman scattering technology relies on unique single-molecule detection capabilities to better understand chemical reactions during friction or lubrication. There was an obvious opportunity to apply SERS and TERS to in-situ tribology, chemical reaction on the friction surface, chemical properties and surface adsorption characteristics of the boundary membrane formed by friction chemistry, and quality detection of ultra-thin thin film. At present, there are still some important problems to be overcome in the development of tribology by SERS and TERS technologies. In the future research, we should take improving the reliability as the main objective to strengthen the research on SERS and TERS in tribology under extreme environment. We believe that SERS and TERS will have more potential in tribology and lubricant detection.

**Author Contributions:** Formal analysis, Z.X.; investigation, K.Z.; writing—original draft preparation, K.Z.; writing—review and editing, K.Z. and A.R.; supervision, Y.S.; project administration, Z.X.; funding acquisition, Z.X., A.R., T.X. and F.F.

**Funding:** The study is supported by National Natural Science Foundation of China (No. 51575389, 51761135106), National Key Research and Development Program of China (2016YFB1102203), State key laboratory of precision measuring technology and instruments (Pilt1705), and the '111' project by the State Administration of Foreign Experts Affairs and the Ministry of Education of China (Grant No. B07014). A. Rosenkranz gratefully acknowledges the financial support given by CONICYT in the framework of the project (Fondecyt 11180121). In addition, Andreas Rosenkranz would like to acknowledge the VID for the financial support given in the project UI 013/2018.

**Conflicts of Interest:** The authors declare no conflict of interest.

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
