# Peer review of "Surface- and Tip-Enhanced Raman Scattering in Tribology and Lubricant Detection—A Prospective"

_lubricants, doi:10.3390/lubricants7090081_

Round 1

Reviewer 1 Report

Authors have conducted a detailed review on application of SERS and TERS techniques in the field of tribology. I have few comments which should be addressed in the review article for a better understanding of audience. Those are listed below:

Lot of time, tribofilm generates in the contact surface, it grows and then gets removed. Eventually it grows again and again gets depleted. My question is, please throw some light on how such changing phenomena can/can’t be captured using this technique.

The authors should discuss a bit about the crucial factors which tribologists should be careful about during their experiments using these techniques to improve the signal quality (S/N ration).

Is there any literature available showing the traceability of such technique? In particular if certain nanoparticle is very minimum in quantity inside the tribofilm, whether this technique can/can’t capture its presence?

Please discuss about the effect of test temperature on lubricant detection capability of these techniques.

Overall, I feel there is a lot written about the great potential of these techniques but very minimum about its challenges. I would suggest writing a paragraph showing the challenges of this technique at current days.

Reviewer 2 Report

Manuscript ID:       lubricants-587311

Title:                     Surface- and tip-enhanced Raman scattering in tribology and lubricant detection - A prospective

Authors:                Kun Zhang, Zongwei Xu, Andreas Rosenkranz, Ying Song, Tao Xue, Fengzhou Fang        

Dear authors,

I think this review helps tribologists to analyze the tribology surfaces by SERS and TERS. Subscribers may ask the differences of these methods from other vibrational spectroscopies such as SFG (sum frequency generation spectroscopy). The spatial resolution of SERS and TERS is of importance. The influence of surface roughness on analytical quality should be addressed.

Round 2

Reviewer 1 Report

The authors have made significant changes to the manuscript, I think it can be accepted in present form.

Reviewer 2 Report

Manuscript ID:       lubricants-587311

Title:                     Surface- and tip-enhanced Raman scattering in tribology and lubricant detection - A prospective

Authors:                Kun Zhang, Zongwei Xu, Andreas Rosenkranz, Ying Song, Tao Xue, Fengzhou Fang        

Dear authors,

Thank you for revised version. In my opinion, it is acceptable.